# A Rigid and Planar Aza-Based Ternary Anhydride for the Preparation of Cross-Linked Polyimide Membrane Displaying High CO_2_/CH_4_ Separation Performance

**DOI:** 10.3390/polym14030389

**Published:** 2022-01-19

**Authors:** Xiaozhou Zhang, Changyu Han, Tianhui Liang, Hongge Jia, Jinyan Wang

**Affiliations:** 1College of Materials Science and Engineering, Qiqihar University, Qiqihar 161006, China; 01727@qqhru.edu.cn (X.Z.); hanchangyu1216@163.com (C.H.); tianhui.liang@foxmail.com (T.L.); zhangxzh-n@163.com (H.J.); 2Heilongjiang Province Key Laboratory of Polymeric Composition Material, Qiqihar University, Qiqihar 161000, China; 3Department of Polymer Science & Engineering, Dalian University of Technology, Dalian 116024, China

**Keywords:** ternary anhydride, hexazobenzene, polyimide, gas separation film

## Abstract

In this study, based on the preparation of hexaazatriphenylene-ternary-anhydride (HAT-T), polyimide membranes were prepared by reaction of 4,4′-(hexafluoroisopropylidene)diphthalic anhydride (6FDA), 4,4′-diaminodiphenyl sulfide (SDA), 2,2′-bis (trifluoromethyl)diaminobiphenyl (TFDB) and 5-amino-2-(4-aminophenyl) benzimidazole (PABZ). Polyimide films with a hexazobenzo structure have good film-forming properties, high molecular weight (*M*_n_ = 0.79–11.79 × 10^6^, *M*_w_ = 1.03–16.60 × 10^6^) and narrow molecular weight distribution (polymer dispersity index = 1.17–1.54). With the introduction of rigid HAT-T, the tensile strength and elongation at break of polyimide films are 195.63–510.37 MPa and 4.00–9.70%, respectively, with excellent mechanical properties. The gas separation performance test shows that hexaazatriphenylene-containing polyimide films have good gas selectivity for CO_2_/CH_4_. In particular, the separation performance of PIc-t (6FDA/PABZ/HAT-T) surpasses the “2008 Robeson Upper Bound”. The selectivity of 188.43 for CO_2_/CH_4_ gas reveals its potential value in the separation and purification of methane gas.

## 1. Introduction

Compared with other separation processes, membrane separation has the advantages of energy saving, less investment, easy installation, light weight, no pollution and easy-to-adjust operation [1,2,3,4]. Gas separation membranes have been widely used in air separation, acid gas separation, gas dehumidification, organic steam recovery and other aspects [5,6]. At present, natural gas has become an indispensable fuel in our life. The exploitation and transportation of natural gas are often accompanied by carbon dioxide (CO_2_). CO_2_ not only affects the quality of natural gas as an impurity, but also corrodes the natural gas pipeline and causes dangerous pipeline leakage over time. Therefore, it is attractive to develop gas separation membranes with excellent performance to separate CO_2_/CH_4_ mixed gas. Polyimide (PI) has been widely investigated in the field of gas separation due to good permeability/selectivity, stable chemical structure, outstanding mechanical properties and high free volume [7,8,9,10,11], Freeman et al. demonstrated that gas selectivity could be further improved by increasing the chain rigidity of the polymer [12]. However, the increasing rigidity of molecular chains will stack molecular chains tightly, thus reducing permeability. A rather general tradeoff relation has been recognized between permeability and selectivity [13,14]. For PI membranes, we found that the high rigidity and large free volume structure in the membrane are conducive to good gas separation performance.

Rigid structures in PIs can be obtained by monomer design or by crosslinking PIs to form networks. The two methods both lead to a decrease in gas penetration flux. Many researchers used hyperbranched structures to construct open channels between chains to improve gas permeability. For example, Jianhua Fang et al. [15] prepared hyperbranched PI using tris (4-aminophenyl) amine (TAPA) and dianhydride monomer by controlling the monomer addition method and monomer mole ratio. Commercially available dianhydride monomers include 4,4′-(hexafluoroisopropylidene)diphthalic anhydride (6FDA), 3,3′,4,4′-diphenylsulfonetetracarboxylic dianhydride and pyromellitic anhydride. By means of a rigid three-dimensional hyperbranched structure, open cavities are formed between adjacent rigid branched chains, and gas permeability is increased while gas selectivity is maintained. With the decrease of ethylene glycol diglycidyl ether crosslinking agent, the permeability flux of CO_2_ and CH_4_ increased by 197.3 and 175%, respectively. However, gelation is difficult to avoid during the preparation of hyperbranched PI. Therefore, strict conditions for polymerization are often required. For example, in the preparation of polyamide acid, a very low monomer concentration must be used, or a slow dripping method is required at low temperatures [16,17,18]. To balance gas permeation flux and selectivity, and prepare high-performance gas separation membranes, many researchers have carried out various explorations. For example, Peter et al. [19] found that when 6FDA was exposed to 4,4′-(hexafluoroisopropylidene)dianiline and 1,3,5-tris (4-aminophenoxy)benzene, the gas permeability coefficient can be significantly increased. This is because the 6FDA comonomer unit contains a large number of –CF_3_ substituents, thus preventing tighter filling of the molecular chain and increasing the free volume of the polymer. Some researchers have also used the combination of rigid and twisted structures to increase the free volume while reducing the activity of the segment. These molecular structures can achieve the goal of increasing both the permeable flux and the gas selectivity [20,21,22,23]. Guoxiong Deng et al. [24] synthesized a series of hyperbranched PIs. They used binaphthalenediamine to provide a spatial-distorting structure, and 1,3,5-tri (4-aminophenyl)benzene to provide a rigid structure of the main chain, which can achieve the goal of balancing the permeability and selectivity of the gas. The gas permeability coefficients of CO_2_ and CH_4_ obtained by them were 73.6 barrer and 1.59 barrer, respectively, and the ideal selectivity of CO_2_/CH_4_ is 46.3, close to the upper limit of 2008 Robeson. Shan Liu et al. [25] used TAPA and 6FDA to prepare PI with microporous network structure. The glass transition temperature of the PI was 308 °C, the gas permeability coefficients of CO_2_ and CH_4_ were 37.4 and 0.66, respectively, and the ideal selectivity of CO_2_/CH_4_ was 56.7, which was close to the upper limit of 2008 Robeson.

In the present studies, a ternary anhydride monomer was used to polymerize with different dianhydride and dihydride amines to construct cross-linked PIs. Different from previous literature, PI films were prepared from hexaazatriphenylene-ternary-anhydride (HAT-T), 6FDA and different diamine monomers in this study. Hexaazatriphenylene framework contains three pyrazine rings in a symmetrical structure. It is a rigid planar conjugated two-dimensional nitrogenous heterocyclic aromatic molecule. It has an electron-deficient structure. In the present study, hexaazatriphenylene derivatives are not only used in the construction of supramolecular systems, the preparation of self-assembled materials, but are also used in the design of gas adsorption, liquid crystal materials and porous organic materials [26]. Xiang Zhu et al. [27] prepared a porous conjugated triazine organic skeleton material by in situ doping. Because of the rich nanopores and the synergistic effect of N/O doping with CO_2_, the material has excellent CO_2_ adsorption properties.

In this work, different diamine monomers (4,4′-thiodianiline (SDA), 2,2′-bis (trifluoromethyl)benzidine (TFDB) and 5-amino-2-(4-aminophenyl) benzimidazole (PABZ)) were used to prepare PI, and the influence of HAT-T on the properties of PI film was studied. The results show that when HAT-T monomer is introduced, the maximum separation coefficient *α* of PI film (PIc-t) can reach 188.43. The results show that the rigid PI with hexaazatriphenylene structure introduced by ternary anhydride is not only easy to prepare but is also a potential gas separation membrane material.

## 2. Experimental

### 2.1. Materials

2,3,6,7,10,11-hexacyano-1,4,5,8,9,12-hexaazatri-phenylene (HAT-CN, 99%), 6FDA (98%), SDA (98%), TFDB (98%) and PABZ (98%) were purchased from Shanghai Aladdin Biochemical Technology Co., Ltd. The reagents N,N-dimethylformamide (DMF), acetic anhydride (Ac_2_O), chloroform (CHCl_3_), acetone, sodium nitrite, sodium hydroxide, sodium bicarbonate, trifluoroacetic acid, acetonitrile, benzene and methanol were obtained from Tianjin Kemiou Chemical Reagents. The reagents were not further purified before use. All test gases were bought from Daqing Xuelong Petrochemical Technology Development Co. Ltd. (Daqing, China) with purities > 99.99%.

### 2.2. Synthesis of Ternary Anhydride

HAT-T was prepared according to the literature [28,29,30]. Figure 1 shows the synthesis route of HAT-T.

#### 2.2.1. 2,3,6,7,10,11-. Hexaacylamino-1,4,5,8,9,12-Hexaazatri-Phenylene (HAAHAT)

HAT-CN (2.4 g, 6.25 mmol) and concentrated sulfuric acid (50 mL) were stirred and dissolved for 72 h. The solution was dropped into stirred water to obtain a yellow suspension, filtered, washed with water (3 × 100 mL), washed with acetone (3 × 100 mL) and then dried in a vacuum.

#### 2.2.2. 2,3,6,7,10,11-. Hexacarboxyl-1,4,5,8,9,12-Hexaazatri-Phenylene (HCBHAT)

Sodium nitrite (3.5 g, 45 mmol) and acetic acid (75 mL) were added in batches within 15 min to the mixed solution of HAAHAT (2.46 g, 5 mmol) and trifluoroacetic acid (TFA) (75 mL), placed in an ice bath, stirred for 12 h, then poured into water and filtered to collect crude products. The solid was dissolved in sodium bicarbonate solution (20 g, 100 mL H_2_O) and treated with sodium hydroxide solution (20 g, 100 mL H_2_O). A yellow precipitate was generated, washed with 50% aqueous ethanol (3 × 50 mL) and vacuum dried at 80 °C. The yellow solid was suspended in 50 mL water, heated to 50 °C, acidified with 50 mL hydrochloric acid, heated at 90 °C for 12 h, filtered, washed with 10% hydrochloric acid (3 × 50 mL), washed with water (3 × 50 mL), filtered and vacuum dried at 80 °C.

#### 2.2.3. Hexaazatriphenylene-Ternary-Anhydride (HAT-T)

HCBHAT (0.625 g, 23.8 mmol) was added to 30 mL acetic anhydride, protected by N_2_, stirred at 115 °C for 1 h, cooled to room temperature, vacuum rotary evaporated to remove the solvent and recrystallized with acetonitrile/benzene.

### 2.3. Synthesis of Hexaazatriphenylene-Containing Polyimides

The PIs PIa–PIc-t of 6FDA and HAT-T, with different diamines (SDA, TFDB and PABZ) were synthesized. The syntheses of PIs with different diamines were similar. Take PIa-t synthesis as an example. A mixed solution of 6FDA (2.2212 g, 5 mmol), HAT-T (15.55 mg, 0.035 mmol), DMF 10 mL and SDA (991.88 mg, 4.953 mmol; 20 mL DMF) solution, N_2_ protection, solid content of 10%, ice bath reaction for 16 h, then 2 mL of acetic anhydride and pyridine mixed solution (V acetic anhydride: V pyridine = 1:1) was added, reacted at 60 °C for 6 h, cooled to room temperature, poured into methanol solution, precipitated to yellow solid, filtered, dried and fibrous PI polymer containing the hexazazobenzophenol structure was obtained. Figure 2 shows the synthesis route of hexaazatriphenylene-containing PIs.

### 2.4. Polyimide Film Preparation

The polymer was dissolved in chloroform at a concentration of 2 wt%, filtered and cast onto a glass plate. The film was further dried at 80.0 °C/1 h, 120.0 °C/1 h, 160.0 °C/1 h and 200.0 °C/1 h to obtain the corresponding hexaazatriphenylene-containing PI film.

### 2.5. Characterization

The Fourier transform infrared (FTIR) spectra of intermediates and monomer powders and the polymer films were measured using a Nicolet NNexus-470 Fourier transform spectrophotometer. ^1^H NMR spectra were obtained on a Bruker Avance 600 spectrometer operating at 600 MHz using DMSO-d_6_ as the solvent. The molecular weight of HAT-T was determined using a Bruker ultrafleXtreme flight mass spectrometer (MS) with acetonitrile as the solvent. The mechanical properties of thin films were tested using an xLW-500N film tensile strength tester from Jinan Sanquan Zhongshi Experimental Instrument Co., Ltd. (Jinan, China). The spline size was 3 cm × 4 cm. Each film was tested three times and the test results were averaged. Molecular weight and molecular weight distribution of the PIs were measured using a Polymer Laboratories PL-GPC120 gel permeation chromatograph (GPC) with polystyrene as an external standard and DMF as the eluent. X-ray diffraction analysis (XRD) of the films was carried out on a Bruker AXS-D8 X-ray diffractometer at 40 kV, 40 mA and Cu-Kα radiation. The *d*-spacing was calculated using Bragg’s equation, nλ=2dsinθ, where *n* is the order of reflection, *λ* is the wavelength of X-ray radiation and *θ* is the XRD angle [31].

### 2.6. Measurements of Gas Transport Properties

The gas permeability of the prepared PI gas separation film was measured using a differential pressure gas permeability meter. The permeability coefficient of a gas (*P*) represents the permeability rate of the permeable membrane. *P* is calculated by the volume of gas passing through the sample per unit thickness and area per unit time (1 barrer = 10^–1^^0^ cm^3^ (STP) cm cm^–2^ s^–1^ cmHg^–1^). The calculation formula is shown in Equation (1):(1)$$P=\frac{q\times k\times L}{A\times p\times t}\times 1000\;(\mathrm{barrer})$$
where *q* transport volume (mL), *k* auxiliary positive coefficient: 2 (fixed value), *L* film thickness (cm), *p* pressure (cmHg), *A* transmittance area: 0.785 (cm^2^) fixed value and *t* test time (s).

The ideal gas displacement selectivity of gas *A* to gas *B* is expressed as the ratio of the permeability coefficient of the former to the latter, which is called the gas selectivity coefficient and is expressed by *α*. The calculation formula is shown in Equation (2).
(2)$$\alpha_{A/B}=\frac{P_{A}}{P_{B}}$$
where *α_A/B_* is the selectivity of the membrane for component *A* over component *B*.

The permeability coefficient, separation coefficient, dissolution coefficient and diffusion coefficient of CO_2_/CH_4_ in this study were obtained by taking the average values of three tests calculated by the formulas listed above.

## 3. Results and Discussion

### 3.1. Characterization of Monomer and Intermediate

The chemical structure of intermediates was confirmed using ^1^H NMR and FTIR. Because there was no hydrogen in the HAT-T structure, time-of-flight mass spectrometry was used to confirm the structure.

Figure 1 shows the FTIR spectra of intermediates. As seen in Figure 1, different from HAT-CN, HAAHAT exhibited the stretching vibrations of C=O and N–H at 1720 cm^−1^ and 3500–3100 cm^−1^, respectively, and the peak of C≡N at 2240 cm^−1^ disappeared, indicating the successfully obtained amide structure. HCBHAT exhibited the stretching vibration of O–H at 3300–2500 cm^−1^; the peak of N–H at 3500–3100 cm^−1^ disappeared, indicating that the amide structure reacted. The stretching vibration of O–H at 3300–2500 cm^−1^ was obviously weakened, indicating the successfully obtained HAT-T.

As seen in Figure 2, peaks (at 8.47 ppm and 8.04 ppm) are assigned to NH_2_ of 2, which is produced by HAT-CN reacting to form 2. Figure 3 shows a peak (at 14.72 ppm) assigned to the hydroxyl structure formed.

The molecular weights of HAT-T were measured using MS. The theoretical calculated value of HAT-T is 444.27, and the experimental test value is 442.1 (Figure 4), which proves that HAT-T is successfully obtained.

### 3.2. Characterization of Hexaazatriphenylene-Containing Polyimide Films

Figure 5 shows the typical FTIR curves of the PIs. The characteristic bands of PIs were found at 725 cm^−1^ (O=C–N–C=O imide ring band deformation), 1720 cm^−1^ (asymmetric C=O stretching vibration) and 1780 cm^−1^ (symmetric C=O stretching vibration) in all spectra. In addition, the stretching vibration of N–H at 3220–3450 cm^−1^, the stretching vibration of CO–NH at 1660 cm^−1^ and the stretching vibration of C–NH at 1550 cm^−1^ disappeared in the PIs’ infrared spectra, indicating that the precursor polyamide acid has been successfully transformed into the expected PI.

Regardless of the addition of HAT-T or not, the PI prepared by chemical imidation in this study is soluble in DMF, so the molecular weight can be determined using GPC. The number-average mass (*M*_n_), weight-average mass (*M*_w_) and polymer dispersity index (PDI) of PIs were tested using GPC with polystyrene as standard material. As seen in Table 1, the PIs obtained in this study have high molecular weight. The PDI ranges between 1.17 and 1.54. Compared with PIa and PIc, PIb has a higher molecular weight. When HAT was added to the preparation of PI, the molecular weight of the obtained polymers remained large. The *M*_w_ of PIb-t was the highest of the PIs at 16.60 × 10^6^.

As the number of C-F per repeating polymer chain remains the same, i.e., the number of fluorine atoms remains the same, the atomic concentrations of other elements are normalized by that of fluorine. Table 1 shows the results of XPS surface analysis of PIa–c-t films. As the HAT-T is introduced into the PIa–c films, the normalized concentrations of C, N and O increase because PIa–c-t films have higher C, N and O contents than PIa–c films.

Figure 6 shows the XRD results for different PI films. The most obvious XRD peak in the spectrum of amorphous polymer is usually used to estimate the average spacing distance (*d*-spacing) in polymer chains. Their corresponding *d*-spacing and 2*θ* values are listed in Table 2. It can be observed that all membranes exhibit two peaks (A and B), which are located at 2*θ* = 16° ± 1° and 22.6° ± 0.2°. The *d*-spacing relative to peak B (*d*-B) of all PIs is very similar. The changes in the *d*-B value are within ±0.1 Å from 3.90 Å. Typical interchain distances of aromatic PIs range from 5.33 to 5.99 Å [32]. Peak A corresponds to the average interchain packing, and wide peaks indicate ordered stacking of PI molecular chains. Peak B can be assigned to the π–π stacking of the imide and phenyl aromatic heterocyclic rings in the polymer backbones, and the spikes indicate that some plane-rigid aromatic heterocycles in the PI skeleton are arranged in parallel [33,34,35]. Regardless of whether the hexaazatriphenylene structure was added to the PI, the chemical structures of imides and phenyl rings do not change. Therefore, the *d*-B value relative to the π–π stacking between them stays more or less the same. When hexaazatriphenylene was introduced into PI, the *d*-A value decreased compared with that without hexaazatriphenylene. In the PI films prepared in this study, the minimum *d*-A value of PIc-t is 5.43 Å. After the introduction of hexaazatriphenylene into the PI, the polymer chains are connected by a planar aza-based group, which significantly decreases the chain freedom. This trend is in qualitative agreement with the gas permeation properties of the polymers, as discussed below.

After adding HAT-T to the monomer, PIa and PIb showed a significant loss in both tensile strength and elongation at break (Table 3). This is because HAT-T is a planar rigid structure. When HAT-T monomer is added, the movement of PIa-t and PIb-t polymer molecular chain segments is restricted and the film becomes brittle, so the mechanical properties are decreased compared with PIa and PIb. Surprisingly, the tensile strength and elongation at break of the PI with the benzimidazole group increased from 472.4 to 510.4 MPa and 3.8 to 9.7%, respectively, when HAT-T is added to the monomer. This is mainly because there are three C=O bonds in the hexaazatriphenylene structure, and it is easier to form hydrogen bonds with the N–H structure contained in the benzimidazole structure of the PIc. In this way, the interaction between macromolecular chains is strengthened and the mechanical properties of PI films are enhanced.

### 3.3. Pure Gas Permeation

As reported in Table 4, the gas separation performance of the six PI membranes (PIa, PIa-t, PIb, PIb-t, PIc and PIc-t) was evaluated using the pure gases of CO_2_ and CH_4_ at 35 °C and 50 atm. For all PI membranes prepared, the order of gas permeability is *P* (CO_2_) greater than *P* (CH_4_). Obviously, this behavior matches the variation in their aerodynamic diameters (CO_2_, 3.3 Å; CH_4_, 3.8 Å) [36]. For CO_2_ or CH_4_, the gas permeability of PIb is the highest because the presence of the –CF_3_ group in the polymer prevents closer filling of the molecular chain, thus improving the free volume of the polymer and increasing the gas permeability. Furthermore, the gas permeabilities of PIa-t and PIb-t are lower than that of the PI film without adding hexaazatriphenylene structure in the chain structure. This is because when adding hexaazatriphenylene structure in the chain structure, the formation of a denser crosslinking structure between the molecular chains reduces the chain spacing. However, for the PI films containing benzimidazole, the gas permeability does not change significantly whether the chain structure contains the hexaazatriphenylene structure or not. This is because imidazole groups, as dissolved groups of CO_2_, can maintain such a high permeability for CO_2_ even though the hexaazatriphenylene structure is introduced as to reduce the interchain distance of the polymer [37]. According to Freeman’s theoretical analysis, the higher the stiffness of the polymer chain, the better the gas selectivity. Therefore, it can be seen from the data in Table 4 that when the hexaazatriphenylene structure is introduced into the polymer, its gas selectivity is increased. In the six types of PI films prepared in this study, the PIc-t had the best gas selectivity, reaching 188.43, surpassing the “2008 Robeson Upper Bound” (Figure 7), and those of other PIs (PIa, PIa-t, PIb, PIb-t and PIc) are positioned near the “1991 Robeson Upper Bound.” This is mainly because of the use of 6FDA as the dianhydride monomer in the preparation of PI; the bulky–(CF_3_)_2_–linkage can better restrict the torsional motion of the two connected phenyl rings. For PIc-t, the polymer chain structure contains both imidazole and hexaazatriphenylene structures. Although the introduction of hexaazatriphenylene into the PI polymer chain causes distance to decrease, the imidazole group has good solubility of CO_2_, and the hexaazatriphenylene structure has excellent absorption of CO_2_. PIc-t thus maintains a high permeability for CO_2_ while increasing selectivity, surpassing the “2008 Robeson Upper Bound.” Robeson upper limit is a commonly used index to evaluate the comprehensive performance of gas separation membranes. Robeson published the Robeson upper bound theory by comparing data on the actual performance of gas separation materials of different structures. Due to the continuous growth of separation membrane technology, more and more gas separation membranes exceeded the Robeson upper bound by 1991. So, Robeson updated the “Upper Bound” in 2008 [38].

Figure 7 also shows a comparison of the gas separation performance of the six PI films synthesized in this study and the PI films prepared using 6FDA in other literature reports. It can be seen from the figure that the structure of the prepared PI monomer plays a crucial role in permeability and separation. Except for 6FDA-MMBDA and 6FDA-DAT1-OH, the other three PIs in Figure 7 all undergo thermal rearrangement at a certain temperature, and the permeability increases based on maintaining a certain selectivity. The selectivity of 6FDA-MMBDA and 6FDA-DAT1-OH polymers is still far from that of PIc-t prepared in this study, although the structures of 6FDA-MMBDA and 6FDA-DAT1-OH polymers have large volume resulting in increased permeability. In this study, considering the preparation of PI membrane, HAT-T was added to the monomer at low content, but it still significantly improved the gas separation performance of the polymer, which also indicated that the plane-rigid structure of hexaazatriphenylene was very effective in improving the gas selectivity. Further studies will be conducted to maximize permeability while maintaining gas selectivity by effective means.

## 4. Conclusions

In this study, PI films (PIa–c-t) were synthesized by the preparation of the HAT-T ternary anhydride monomer containing the hexaazatriphenylene structure, the commercial dianhydride 6FDA and commercial diamines (SDA, TFDB and PABZ). The experimental results show that the PIs with the addition of hexaazatriphenylene structure have high molecular weight and good mechanical properties. In addition, the introduction of the hexaazatriphenylene structure has an important effect on the permeability and selectivity of a gas. Compared with the PI film without the hexaazatriphenylene structure, the gas permeability of the PI film with the introduction of the hexaazatriphenylene structure decreased, but the gas selectivity increased. The selectivity for CO_2_/CH_4_ gas is 188.43, which surpasses the “2008 Robeson Upper Bound”. This is because the introduction of the hexaazatriphenylene structure can effectively reduce the interchain interaction and inhibit chain stacking. The rigid planar structure of hexaazatriphenylene can also improve the selectivity for CO_2_ gas. Therefore, this study is of great significance for the design and synthesis of cross-linked PIs and their application in gas separation membranes.

## Data Availability

Not applicable.

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
