# Peer review of "A Rigid and Planar Aza-Based Ternary Anhydride for the Preparation of Cross-Linked Polyimide Membrane Displaying High CO2/CH4 Separation Performance"

_polymers, 2022, doi:10.3390/polym14030389_

Round 1

Reviewer 1 Report

This is an interesting work about the preparation of polyimides (PI) by addition of hexaazatriphenylene- ternary-anhydride, and the effect on the selectivity for gas separation,. Just a few minor issues to be addressed:

  • 1) Row 133: at this point, it is not clear what do PIa-t and so on represent. I suggest the authors to add a table to better explain the sample labeling (even if reported in scheme 2). In particular, it should be emphasized that a, b and c represent the different diamines, and t is the reaction carried with HAT-T
  • 2) there should be an equality sign between the P ration and D ratio
  • 3) Discussion to FTIR: why samples are coded as 1,2,3,4 ?
  • 4) Table 4: the results for D and S are not even discussed, so for a better reading of the table I suggest the authors to remove the data and also the corresponding equations
  • 5) Figure 7: it is better to explain the difference between the 1991 and 2008 upper bounds,a dn explain how were the curves obtained

Author Response

Dear editor,thank you for your review and comments. Here are my answers.

Reviewer 2 Report

In this paper, the authors describe that the preparation of polyimides from a ternary anhydride, a dianhydride and diamines, and the gas separation performance of their films. This paper is meaningful for the development of gas purification membranes. I recommend that this paper is published after minor revisions according to following comments.

1) Lines 16 and 19

It is better not to use abbreviations (HAT-T and PIc-t) in the abstract.

2) Line 210-

The content of HAT-T is a very important value in this study. It may be difficult to estimate, but the authors need to calculate the content of HAT-T units in the PI polymers from elemental analysis or 1H-NMR data.

3) Lines 268-271

This part is the caption of the table. The text format is used.

Author Response

(The authors gave the same response as above.)

Reviewer 3 Report

The manuscript presents results on the synthesis of various polyimide films and their properties in the gas separation for methane/carbon dioxide separation. The research fits the profile of the journal. The manuscript is well written, however Authors should address the following comments before the acceptance:
1. Line 88 - avoid using "etc." as it does not bring any knowledge nor valuable information.
2. the abbreviations of the prepared polymers should be better explained (in the manuscript - it is only a part of Scheme 2).
3. Equation 2 should be rewritten, there is an error (i.e. "=" is missing); the discussion of this equation must be provided. However, α is not "separation coefficient" but selectivity coefficient. Please refer to paper by Baker et al. (JMS 348 /2010/ 346). 
4. Moreover, Eq. 3 must be also discussed and explained in detail. Moreover, the equipment used for the determination of diffusion coefficients by the time lag (not "lag time") methods must be shown and exemples of the results must be shown.
5. Authors described the preparation of membranes as "film painting" - how the readers should understand that? There are a number of methods of the preparation of dense films by (probably) phase inversion method, but the "film painting" was never described.
6. Why films were of so much different thicknesses (26-60 μm)? The different thicknesses can influence the gas permeability and membrane selectivity.
7. From the Table 4 it is clear that the relatively high selectivity comes from the much higher sorption of carbon dioxide over the sorption of methane. The diffusivity selectivity is around 1. This fact should be also discussed. How these observations would influence the real selectivity of membranes? Did Authors perform the separation using the binary gas mixture (e.g. for the most promising membrane)?

Author Response

(The authors gave the same response as above.)
